# CoCrFeMnNi$_{0.8}$V/Cr$_3$C$_2$-Ni$_{20}$Cr High-Entropy Alloy Composite Thermal Spray Coating: Comparison with Monolithic CoCrFeMnNi$_{0.8}$V and Cr$_3$C$_2$-Ni$_{20}$Cr Coatings

**Stavros Kiape** [1,*], **Maria Glava** [1], **Emmanuel Georgatis** [1], **Spyros Kamnis** [2,3], **Theodore E. Matikas** [1] **and Alexandros E. Karantzalis** [1]

1  Department of Materials Science and Engineering, University of Ioannina, 45110 Ioannina, Greece; glavamaria00@gmail.com (M.G.); mgeorgat@uoi.gr (E.G.); matikas@uoi.gr (T.E.M.); akarantz@uoi.gr (A.E.K.)

2  Computer Science and Biomedical Informatics, University of Thessaly, 35131 Lamia, Greece; spyros.kamnis@castolin.com

3  Castolin Eutectic-Monitor Coatings Ltd., Newcastle upon Tyne NE29 8SE, UK

*  Correspondence: s.kiape@uoi.gr

**Abstract:** High-entropy alloys (HEAs) are revolutionizing the field of surface engineering, challenging traditional alloy frameworks with their superior mechanical attributes and resistance to corrosion. This investigation delves into the properties of the CoCrFeMnNi$_{0.8}$V HEAs, both as a standalone material and when blended with Cr$_3$C$_2$-Ni$_{20}$Cr, to evaluate their efficacy as cutting-edge surface treatments. The addition of vanadium to the CoCrFeMnNi$_{0.8}$V alloy results in a distinctive microstructure that improves hardness and resistance to abrasion. The incorporation of Cr$_3$C$_2$-Ni$_{20}$Cr particles enhances the alloy's toughness and longevity. Employing high-velocity oxy-fuel (HVOF) thermal spray methods, these coatings are deposited onto steel substrates and undergo detailed evaluations of their microstructural characteristics, abrasion, and corrosion resistance. Findings reveal the CoCrFeMnNi$_{0.8}$V coating's exceptional ability to withstand corrosion, especially in environments rich in chlorides. The hybrid coating benefits from the combination of the HEA's inherent corrosion resistance and the enhanced wear and corrosion resistance provided by Cr$_3$C$_2$-Ni$_{20}$Cr, delivering comprehensive performance for high-stress applications. Through the fine-tuning of the application process, the Cr$_3$C$_2$-Ni$_{20}$Cr reinforced high-entropy alloy coating emerges as a significant advancement in protective surface technology, particularly for use in marine and corrosive settings. This study not only highlights the adaptability of HEAs in surface engineering but also prompts further investigation into innovative material pairings.

**Keywords:** high entropy alloys; thermal spray coatings; microstructure; wear; corrosion

## 1. Introduction

### 1.1. High-Entropy Alloys

High-entropy alloys (HEAs) represent a groundbreaking advancement in the field of materials science, challenging the norms of traditional alloy composition. These materials are characterized by their vast compositional diversity, typically featuring five or more main elements in equal or nearly equal proportions. Such unique composition leads to a significant increase in configurational entropy, which is a defining characteristic of HEAs [1,2]. In contrast to traditional alloys, which are based on one primary element with minor additions for property modifications, HEAs embrace a complex mixture that often results in simple solid-solution phases such as face-centered cubic (FCC), body-centered cubic (BCC), or hexagonal close-packed (HCP) structures. The entropic mechanisms underlying HEAs are responsible for their remarkable attributes, including enhanced strength, improved wear and corrosion resistance, and exceptional thermal stability, making them suitable for a wide range of uses [3,4].

The concept of HEAs emerged from foundational research published independently in 2004, introducing this innovative class of materials to the world. These initial studies demonstrated the feasibility of synthesizing single-phase materials with multiple principal elements, establishing HEAs as a significant shift in materials science [1,2]. Further investigations into HEAs have delved into their phase behavior, microstructural characteristics, and ways to optimize their properties. The tendency of HEAs to form solid-solution phases is largely due to the high entropy effect, which, when combined with factors like lattice distortion, slow diffusion, and the cocktail effect (arising from the variance in atomic sizes, melting points, and crystal structures of the elements involved), endows HEAs with unique structural and performance features [5].

Recent research has expanded into the development of multiphase HEAs, which combine various mechanical properties to achieve an optimal balance of strength and ductility. This area of study utilizes a combination of empirical guidelines, phase diagram predictions, and cutting-edge machine learning algorithms [6]. Machine learning, in particular, has become a crucial method for predicting the phases and properties of HEAs, adeptly managing complex data and showing potential in predicting the phases and mechanical properties of HEAs. However, the application of machine learning in HEA development is faced with hurdles, including the need for detailed feature descriptions and the integration of manufacturing parameters, such as cooling rates in thermal spraying processes [7].

Within the scope of surface engineering and coatings, HEAs offer novel alternatives to traditional materials that might be limited, harmful, or restricted. Their flexibility for tailored applications positions them as highly desirable for surface coating technologies, where understanding the impact of rapid solidification and cooling rates is essential [8].

### 1.2. The Demand for Novel Alloys in Surface Engineering

Cemented carbides, renowned for their exceptional hardness and resistance to wear, are a staple in surface engineering applications across numerous sectors. However, they face considerable challenges that necessitate the search for alternative materials. One of the main issues is their high cost, attributed to the expensive raw materials and complex production methods required, which significantly increase overall costs [9]. Furthermore, the supply of these materials is often concentrated in specific geographic areas, making them vulnerable to geopolitical tensions and market volatility. This concentration can lead to supply disruptions, price volatility, and reliance on particular regions for critical resources [10]. Additionally, cemented carbides have been in use for over three decades, and while they have proven effective in many applications, the advancing requirements of modern technologies and industries frequently exceed what these traditional materials can offer. This gap highlights a need for coatings that can perform in more demanding environments, especially where enhanced properties like superior corrosion resistance are crucial [11].

In this context, high-entropy alloys (HEAs) stand out as a viable solution, either on their own or in conjunction with cemented carbides for coating applications. HEAs are particularly attractive for their potential to demonstrate unparalleled hardness, wear, and corrosion resistance, which could surpass that of conventional cemented carbides [12–14]. A distinctive benefit of HEAs is their compositional versatility, allowing for customization from a wide range of elements [15]. This flexibility reduces dependence on costly, scarce, or geopolitically sensitive materials, thus potentially offering more stable prices and supply chains. This advantage is increasingly important in today's geopolitical climate, which can significantly affect material availability and costs [10].

HEAs allow for the creation of alloys specifically designed to meet the requirements of modern applications, unlike traditional cemented carbides. They can be engineered to offer a combination of high hardness, superior wear resistance, and exceptional corrosion resistance, making them suitable for broader applications, including those subject to harsh environmental conditions or extreme stress [16]. As coating materials, HEAs provide an

opportunity to overcome the limitations associated with the long-standing use of cemented carbides. Developing HEAs represents more than just the introduction of a new class of materials; it addresses the complex challenges confronting industries that have traditionally relied on cemented carbides, offering a pathway to innovative and more resilient solutions. In addition to their inherent properties, by strategically combining HEAs with carbides, we unlock a new realm of possibilities in surface engineering and coatings. This unique amalgamation not only enhances the mechanical and thermal properties of the resulting composite but also augments its resistance to wear, corrosion, and thermal degradation. As we delve deeper into the realm of material science, understanding the intricate interplay between HEAs and carbides unveils a promising avenue for the development of next-generation surface coating technologies with superior performance and durability.

### 1.3. The Reasoning behind this Endeavor

The choice of $CoCrFeMnNi_{0.8}V$ as the primary alloy in the development of new high-entropy alloy (HEA) compositions, designed to exhibit enhanced hardness, wear, and corrosion resistance, is grounded in machine learning insights, theoretical analyses, and empirical evidence [17]. Machine learning techniques have pinpointed the integration of Vanadium (V) into the CoCrFeMnNi framework as a pivotal enhancement, particularly when V is added in conjunction with a specific Nickel (0.8) concentration, to amplify certain alloy characteristics [18]. This analytical approach leverages complex data sets to forecast optimum alloy compositions that are likely to achieve the targeted properties.

Experimental investigations have uncovered the fact that the $CoCrFeMnNi_{0.8}V$ alloy, when cast at a cooling rate of around 100 K/s, exhibits a unique microstructure. This structure includes a primary globular sigma phase amidst an FCC matrix, along with V-enriched particles, marking a distinctive phase composition not previously reported for the CoCrFeMnNiV system [19]. Such a microstructural arrangement significantly elevates the alloy's hardness and resistance to wear compared to standard CoCrFeMnNi mixtures. From a thermodynamic perspective, the emergence of V-rich particles and the microstructural development within $CoCrFeMnNi_{0.8}V$ are steered by the mixing heats of various elemental pairs. Notably, the V-Ni combination demonstrates a more favorable mixing heat relative to other pairings such as V-Cr, V-Fe, and V-Mn, indicating a natural tendency for these elemental interactions within the alloy [20].

The alloy's corrosion resistance is particularly attributed to its compositional strategy, especially the inclusion of chromium (Cr), which is renowned for its corrosion mitigation capabilities. Furthermore, vanadium's (V) role in the $CoCrFeMnNi_{0.8}V$ mix is anticipated to significantly bolster corrosion resistance through enhanced passivation effects and the stabilization of the passive film [21,22]. Incorporating CrC-NiCr with the $CoCrFeMnNi_{0.8}V$ HEA is a deliberate move to improve the composite material's overall attributes while addressing CrC-NiCr's inherent brittleness and its potential lower corrosion resistance [23–25]. The integration of $CoCrFeMnNi_{0.8}V$ into the composite not only augments mechanical properties and toughness but also elevates corrosion resistance.

This study aims to empirically validate the augmented performance resulting from merging specific HEAs with CrC-NiCr. The goal is to broaden the scope of applications and surmount existing challenges in critical sectors that demand advanced surface coating solutions, thereby pushing the boundaries of material science in surface engineering.

## 2. Materials and Methods

### 2.1. Feedstock Materials

The high-entropy alloy (HEA) feedstock, $CoCrFeMnNi_{0.8}V$, (density 7.68 g/cm$^3$, melting point $\approx$ 1570 °C) in powder form is a research and development item from Castolin Eutectic, primarily comprising a singular BCC solid-solution phase [15]. This powder has a particle size ranging from 20 to 50 μm and is noted for its lack of porosity and minimal imperfections (microcracks, voids, inclusions, etc.). It is fabricated using the gas atomization technique, which generally entails melting the substance followed by its swift

solidification using a gas stream to create small spherical-like particles. The powder's chemical makeup is nearly equivalent to the system's nominal composition, albeit with a marginally reduced vanadium level and an increased chromium level. Elemental mapping (referenced in Figure 1) showcases an even distribution of these elements [17]. The specific details of the powder's composition are listed in Table 1.

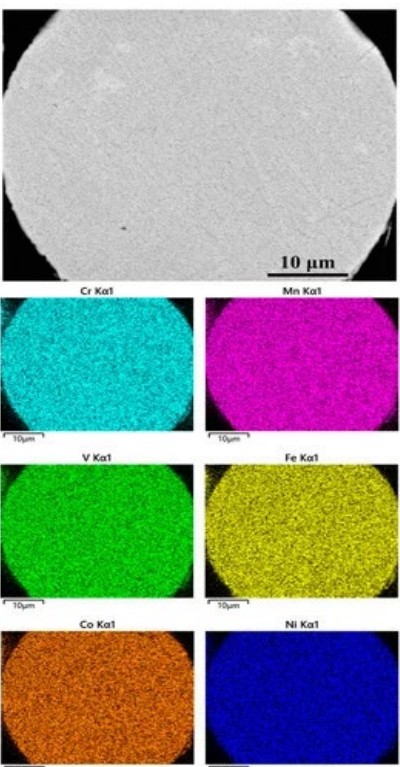

**Figure 1.** Microstructure (BSE mode) and elemental map of CoCrFeMnNi$_{0.8}$V powder, Ref. [17].

**Table 1.** Chemical composition (at%) of gas-atomized CoCrFeMnNi0.8V powder [17].

|  | V | Cr | Mn | Fe | Co | Ni |
|---|---|---|---|---|---|---|
| Nominal | 17.2 | 17.2 | 17.2 | 17.2 | 17.2 | 13.8 |
| Powder | 15.9 ± 0.2 | 18.1 ± 0.2 | 17.9 ± 0.4 | 17.1 ± 0.4 | 17.3 ± 0.2 | 13.8 ± 0.3 |

Figure 2 shows the SEM micrographs of the Cr3C$_2$-NiCr powder. The powder comprises spherical, agglomerated, and sintered powders specifically designed for thermal spraying, consisting of 80% chromium carbide (density 6.68 g/cm$^3$, melting point $\approx$ 1620 °C) for enhanced hardness and resistance to wear, complemented by a nickel–20% chromium matrix that serves multiple functions. This NiCr matrix (density 7.49 g/cm$^3$, melting point $\approx$ 1440 °C) not only prevents the chromium carbide from decomposing (decarburization) during thermal spraying, but also acts as an adhesive for the carbides during use. Upon application via the HVOF (high-velocity oxy-fuel) spray process, the resultant coatings are notably dense, exhibit robust bond strength, and present greater uniformity when compared to coatings applied through plasma spraying or combustion powder techniques [26]. In the top micrograph, a cluster of spherical particles is visible, uniform in shape and with a rough surface texture. These particles are densely packed together, and their size is relatively consistent throughout the field of view. The nominal size is −45 μm (D95) +15 μm (D5). The bottom image is a cross-sectional view of the CrC-NiCr powders used for thermal spraying. The interior of the particles in the bottom image displays a heterogeneous texture, which indicates variations in the material composition due to the presence of Ni-Cr.

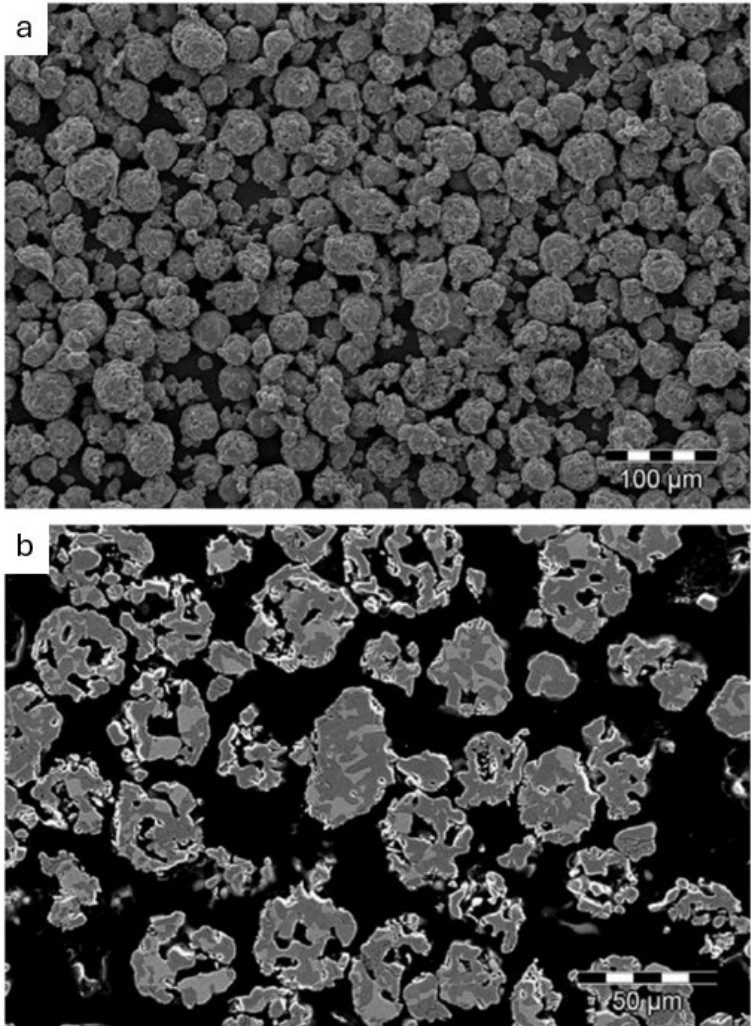

**Figure 2.** SEM micrographs showing the morphology (**a**) and the microstructure (**b**) of the $Cr_3C_2$-20($Ni_{20}Cr$) powder.

### 2.2. Coating Application

Coating trials were conducted on steel plates (SAE 1070) measuring $160 \times 80 \times 3 \text{ mm}^3$. The experiment involved the application of three different coating materials: a pure high-entropy alloy (HEA) powder, a CrC-NiCr powder, and a composite mixture containing 75% HEA and 25% CrC-NiCr by mass. The steel plates were roughened by abrasive blasting with 46 μm alumina particles from a 100 mm distance, followed by a thorough cleaning using high-pressure air and mechanical means to remove any adhering grit from the surface.

The coatings were applied with a proprietary "compact High Velocity Oxy-Fuel (HVOF)" technique [27], featuring a nozzle designed to accelerate exhaust gases to almost trice the speed of sound (Mach 2.7). The in-house optimization of the HVOF gun's parameters focused on producing the ideal microstructure, characterized by low porosity and reduced mean free path of the binder, while maximizing the volume of Cr-C and achieving maximum microhardness for the coatings. The optimization also aimed to minimize decarburization by lowering the flame temperature and maximizing particle impact velocity.

### 2.3. Structural Analysis

The microstructure was analyzed using a JEOL 6510 LV scanning electron microscope, equipped with Oxford Instruments' backscattered electron (BSE) and energy dispersive spectroscopy (EDS) detectors.

### 2.4. Phase Identification

Phase compositions were identified through X-ray diffraction (XRD) analysis, using a Bruker D8 Advance diffractometer (Billerica, MA, USA), with Cu K$\alpha$ radiation over a 2$\theta$ range from 10 $^\circ$ to 120 $^\circ$ at a scan rate of 0.01 $^\circ$/s.

### 2.5. Mechanical Property Assessment

A universal hardness tester (INNOVATEST IN-700) with a Vickers diamond indenter was used to assess hardness under a load of 306 N. Average values and standard deviations were calculated from at least ten measurements per sample.

### 2.6. Tribological Evaluation

Wear resistance was evaluated using a ball-on-disk tribometer (CSM Instruments, Needham, MA, USA), with a 5 N load and a linear speed of 10 cm/s against an $Al_2O_3$ ball with a 6 mm diameter. The tests covered a total distance of 2000 m, with specimen cleaning, weighing, and debris collection occurring every 200 m. To ensure data reliability, each alloy was tested at least three times. Wear rate and mass loss were subsequently calculated.

### 2.7. Electrochemical Stability Testing

Corrosion behavior was assessed via cyclic polarization in a 3.5 wt.% NaCl solution, using a Gamry Reference 600 potentiostat/galvanostat with a three-electrode cell setup. The sample served as the working electrode, a graphite plate as the counter electrode, and a saturated calomel electrode (SCE) as the reference. Prior to the polarization tests, specimens were immersed for an hour to stabilize the open circuit potential, followed by scanning at a rate of 10 mV/min. The pH was measured before and after the test and was found unchanged and close to a value of 7.

## 3. Results and Discussions

### 3.1. Microstructure of the $Cr_3C_2$-$Ni_{20}Cr$ Coating

Figure 3 displays a cross-sectional view of the applied coating. The areas with a lighter appearance represent the nickel alloy binder, whereas the darker grey areas are the grains of chromium carbide. The coating achieved an average thickness of 500–600 μm. The deposition quality of the coating is remarkable, exhibiting an exceptional phase distribution with high consistency and cohesion across the successive layers. In Figure 3b, the presence of bright spots suggests the existence of some porosity within the carbide phase [28–30].

The complexity of the mictostructure of this system is evident in Figure 4. Furthermore, the main phases of the coating are the $Ni_{20}Cr$ alloy phase and the $Cr_3C_2$ carbide phase. As an adhesive phase, the $Ni_{20}Cr$ phase provides a higher bonding strength and toughness, while the carbide phase, which acted as the hard phase, has a higher hardness and wear resistance [31].

### 3.2. Microstructure of the $CoCrFeMnNi_{0.8}V$ HEA Coating

Figure 5a presents a SEM image showing a cross-section of the $CoCrFeMnNi_{0.8}V$ coating. Its lamellar structure is typical of thermally sprayed coatings, comprising layers or 'splats' that are created when molten or semi-molten particles impact and flatten against the substrate. This indicates that the HVOF technique effectively softened and propelled the particles, resulting in the creation of overlapping thin layers that enhance the coating's density and mechanical interlocking, thereby improving fracture toughness and cohesion [32].

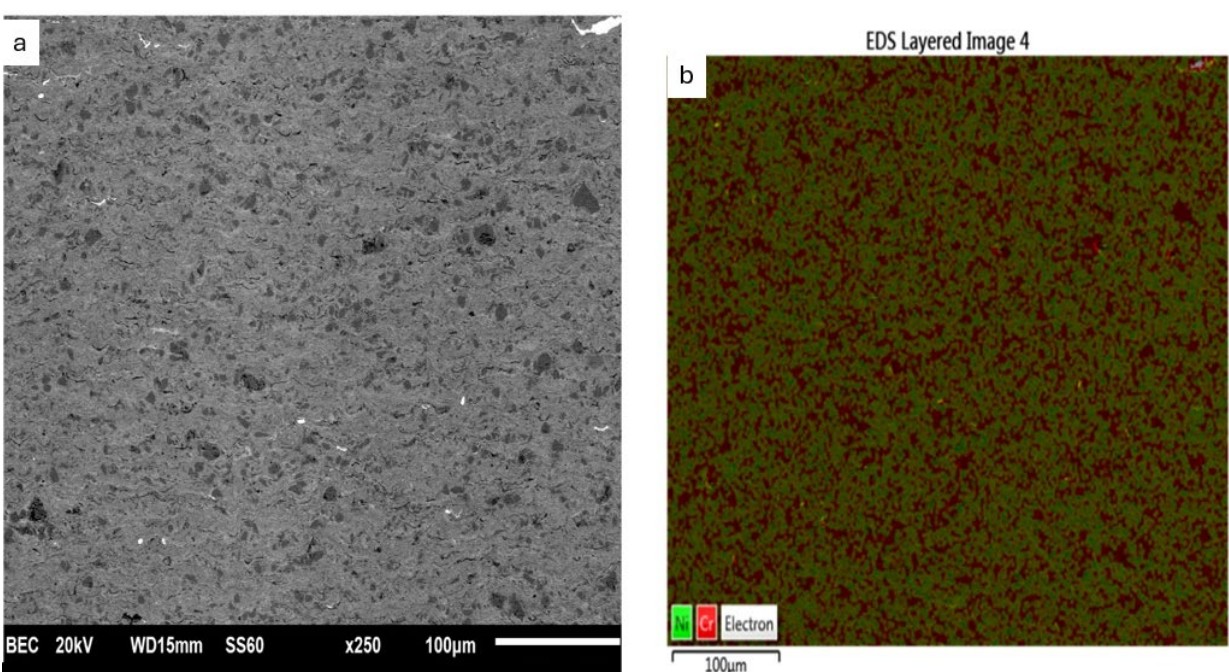

**Figure 3.** (**a**) CrC-NiCr coating microstructure, (**b**) EDX of coating cross-section.

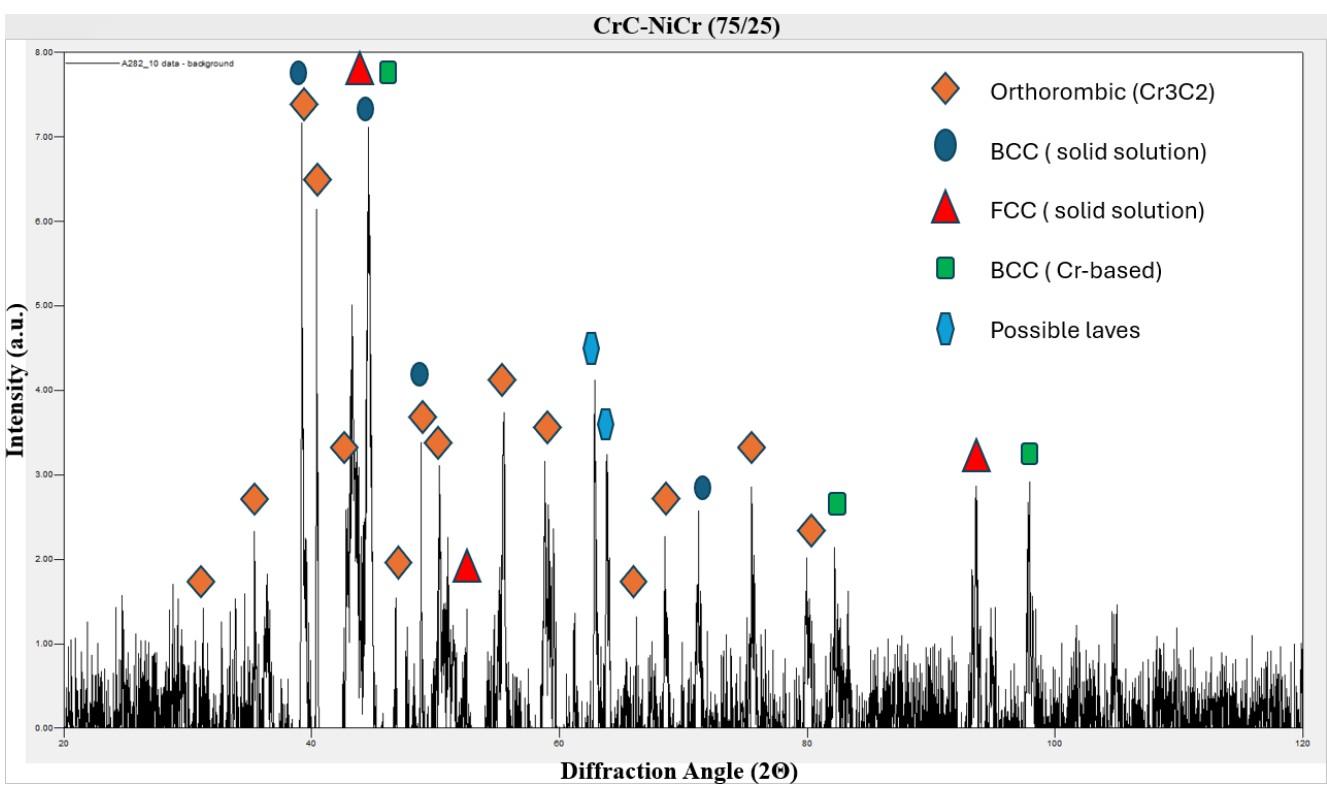

**Figure 4.** X-ray diffraction (XRD) of the sprayed coating.

Oxidation is noticeable at the periphery of the particles, a by-product of the hot particles' short interaction with atmospheric oxygen during spraying. This oxidation at the splat boundaries is a familiar occurrence in HVOF-applied coatings, due to the high-temperature nature of the process [33]. The evident plastic deformation of the particles within the SEM image suggests material ductility, a crucial factor for achieving a dense

coating, as it allows the particles to spread upon impact and form a unified layer with strong mechanical links between splats [34].

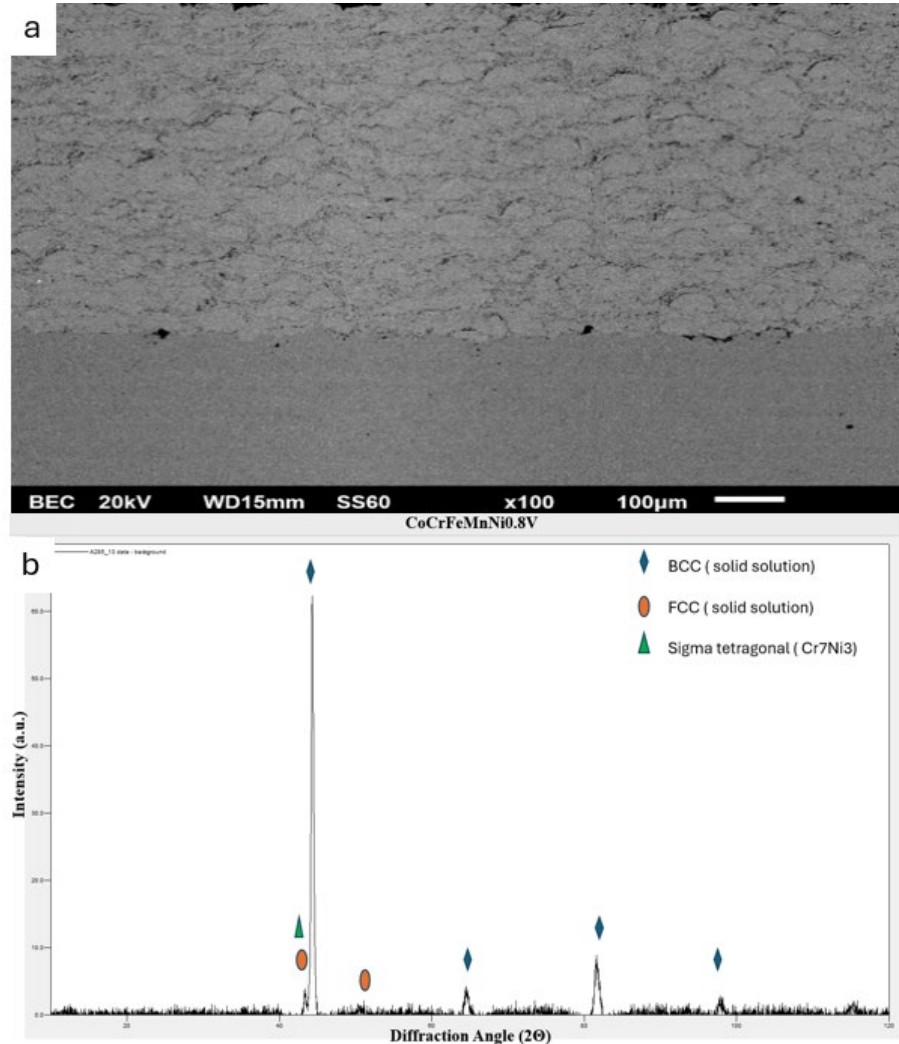

**Figure 5.** (**a**) CoCrFeMnNi$_{0.8}$V coating microstructure, (**b**) XRD of coated material.

The SEM image also reveals a minimal presence of voids or gaps, pointing to a low porosity level within the coating, which is beneficial for enhancing wear resistance and protection against corrosive substances. The clean and uninterrupted interface between the coating and the substrate, free of delamination or significant voids, implies robust adhesion, likely due to the high kinetic energy of the particles upon impact during the HVOF process, which fosters strong mechanical bonding to the substrate [35].

In Figure 5b, the detection of additional phases not found in the original BCC structure of the feedstock material can be ascribed to the thermal spray process. This process involves rapid heating and cooling as the material is deposited onto the substrate, which can lead to phase-composition variations between the coating and the feedstock. The thermal dynamics experienced during spraying may trigger the emergence of new phases or maintain the original feedstock crystal structure in larger particles that do not fully melt. Conversely, smaller particles that melt completely in-flight and resolidify upon contact with the substrate may cool at a slower rate, potentially leading to FCC structures or the formation of secondary intermetallic phases [15].

### 3.3. Microstructure of the Composite Coating

Figure 6a,b show that the coatings are dense, without any visible cracks, indicating strong adhesion to the substrate. This lack of cracking or delamination at the interface is a clear testament to the coatings' structural integrity. The measured thicknesses of the composite coating was withing the range of 450–500 μm. Within the microstructure of the coating, the presence of splats from the HEA alloy and $Cr_3C_2$-$Ni_{20}Cr$ particles was noticeable. Intriguingly, many of the HEA alloy particles maintained their spherical shape/elliptical shape, which is an indication of a partial or negligible melting stage. The average diameter of these spherical HEA alloy splats, based on the measurement of 10 such splats from cross-sectional BSE images, was found to be roughly 40 to 50 μm. It is important to note that these measurements could vary depending on the sample preparation process.

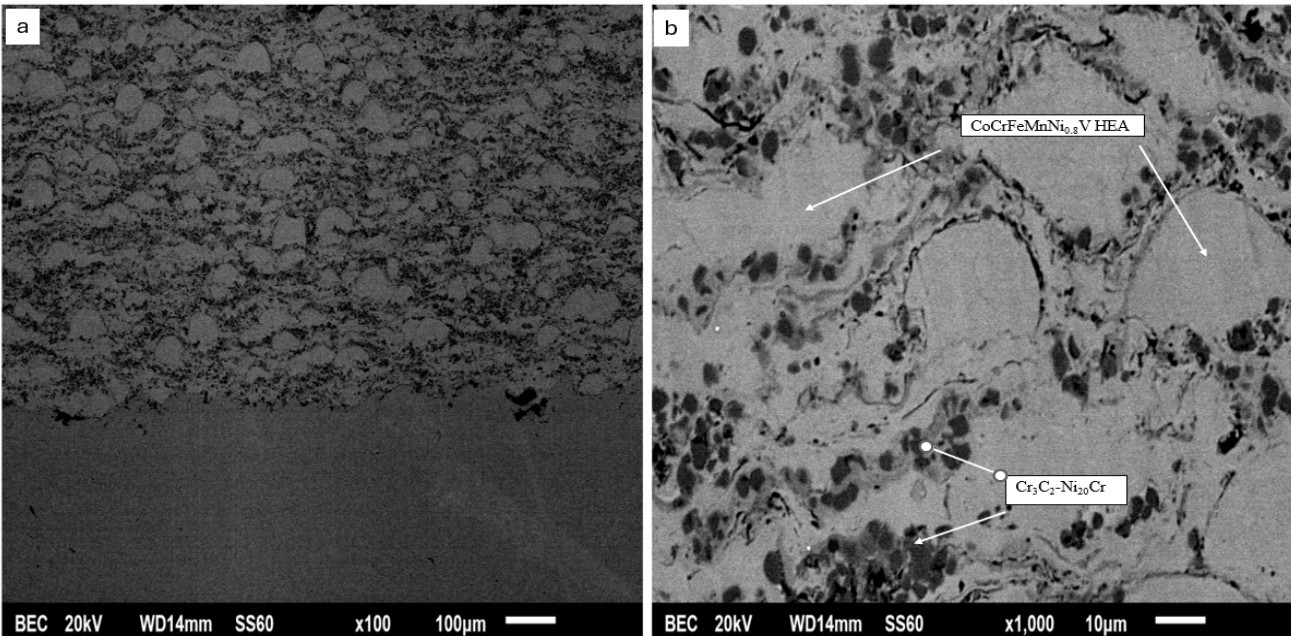

**Figure 6.** (**a**) Composite coating cross-section, (**b**) higher magnification with the involved phases.

Expanding further, X-ray diffraction (XRD) patterns displayed in Figure 7 provided a more in-depth look at the coatings. The sharp Cr-C peaks confirmed the existence of a well-defined carbide phase. The detection of both FCC and BCC phases reflects a complicated solidification process and phase stability that are dependent on the cooling rate experienced during spraying. The presence of decarburized phases, denoted by green markers at specific angles, suggests the development of amorphous or nanocrystalline structures during the spray procedure. Additionally, the peaks corresponding to minor phases in the HEA alloy appeared diminished in intensity in the sprayed coatings compared to the powder feedstock. This change could be attributed to these minor crystalline phases dissolving into a solid solution amidst the spray process, which lacked enough time to fully recrystallize upon quenching as the particles impacted the substrate.

### 3.4. Sliding Wear Response

3.4.1. $Cr_3C_2$-$Ni_{20}Cr$ Coating

Figure 8 shows the mass loss as a function of the sliding distance. From the slope of these curves, and taking into consideration the externally applied load, the wear rate was calculated (Table 2).

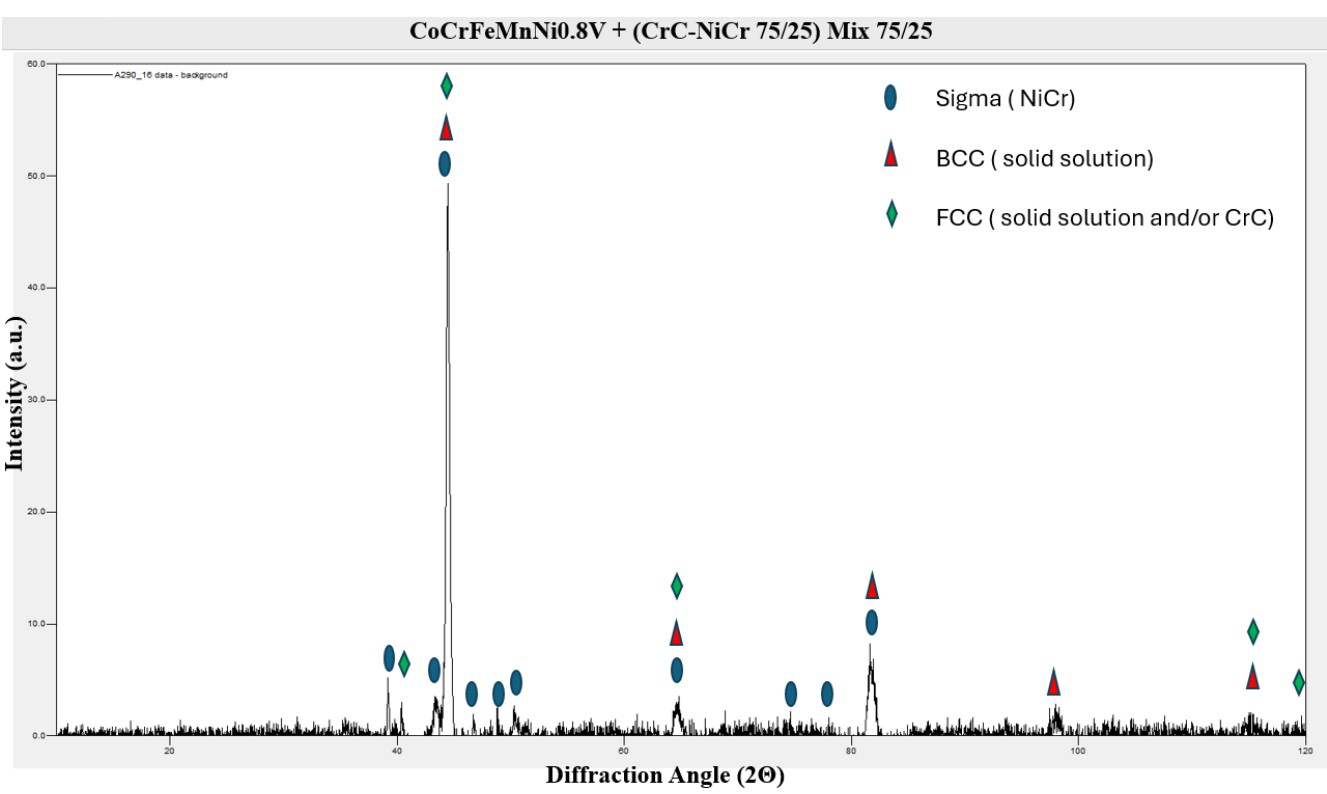

**Figure 7.** X-ray diffraction (XRD) of the sprayed coating.

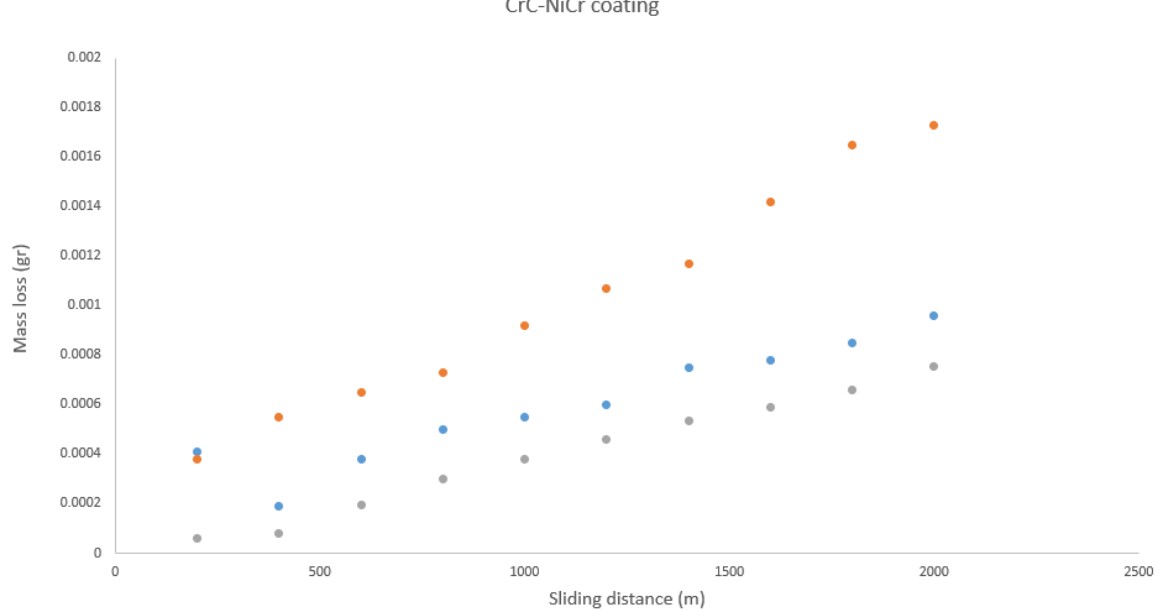

**Figure 8.** Mass loss as a function of sliding distance for the $Cr_3C_2$-$Ni_{20}Cr$ coating.

Figure 9a–c displays SEM images from the wear test of a $Cr_3C_2$-$Ni_{20}Cr$-coated surface. The surface exhibits a coarse texture characterized by channels and abrasions, symptomatic of abrasive wear mechanisms. This form of wear likely results from the action of hard particles either rolling across or dragging along the coating, predominantly stripping material from the comparatively less hard Ni-Cr matrix that embeds the harder CrC granules [36].

**Table 2.** Average Specific Wear rates for the different coating materials.

| Coating System | Average Specific Wear Rate ($\times 10^{-7}$ g/Nm) | Coefficient of Friction | Hardness HV | Wear Mechanism |
|---|---|---|---|---|
| CrC-NiCr | 1.12 | 0.493 | 893 | Abrasive + Fatigue |
| CoCrFeMnNi$_{0.8}$V | 7.13 | 0.595 | 500 | General |
| Composite (75%HEA-25%Cr3C2-Ni20Cr) | 1.70 | 0.654 | 540 | Combined |

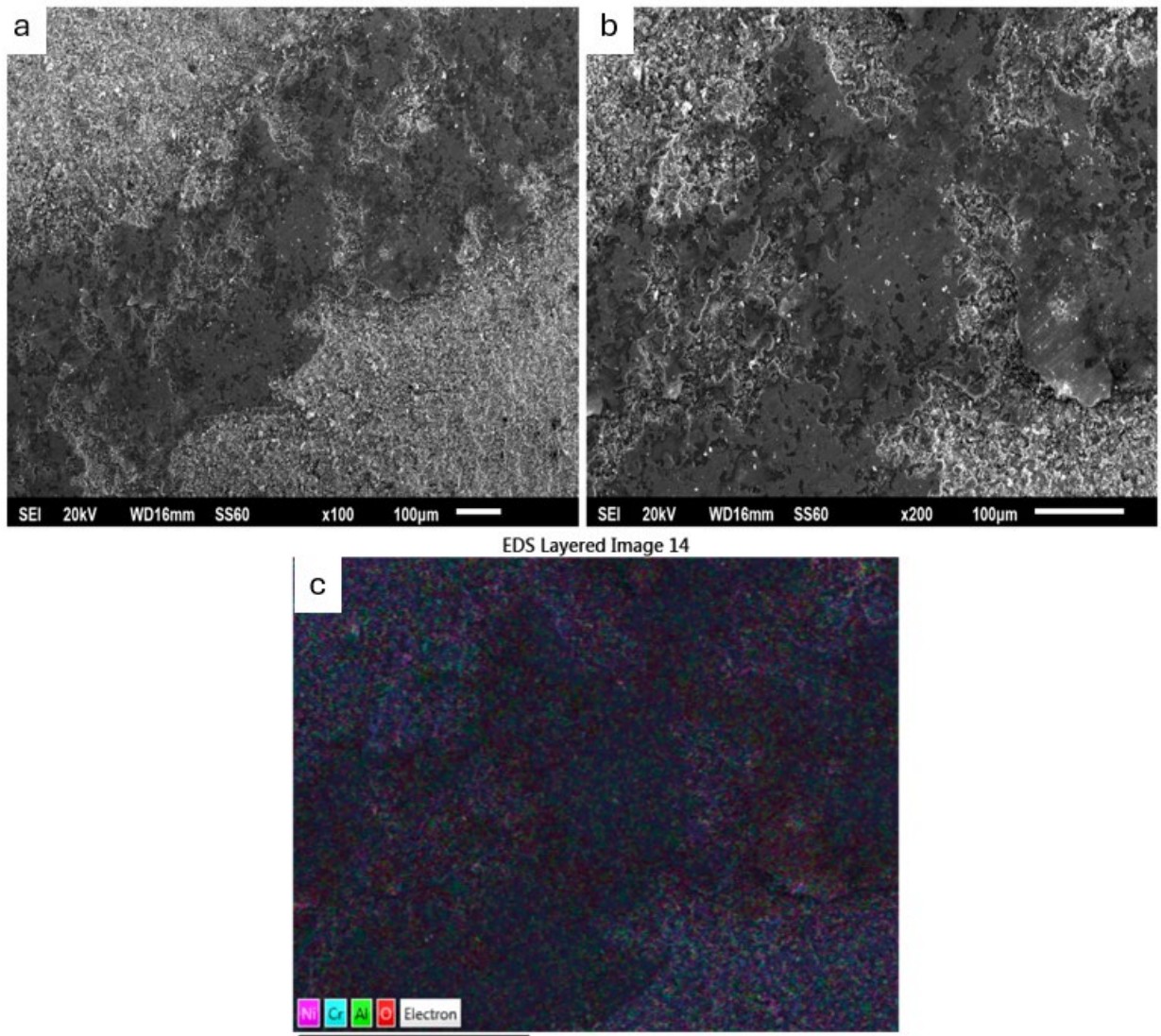

**Figure 9.** (**a**,**b**) Wear tracks of $Cr_3C_2$-$Ni_{20}Cr$ coating at different magnifications, (**c**) EDS of wear track.

Elemental composition analysis via energy-dispersive X-ray spectroscopy (EDS) (Figure 9c) sheds light on the wear track's elemental makeup. The detection of aluminum and calcium—elements extrinsic to the CrC-NiCr composition—hints at the contribution of foreign materials in the wear process. The aluminum could point to material transfer from a contact surface or the participation of extraneous abrasive entities.

A lower-magnification SEM image (Figure 9a) offers a panoramic view of the wear path, revealing varying contrast zones likely linked to distinct wear phases or the incorporation of debris within the wear groove. The heterogeneous wear pattern may indicate varying wear severities or different wear actions across the track. The brighter regions,

generally associated with the sturdier CrC particles, imply that these zones have withstood the wear, whereas the encasing darker matrix—the NiCr component—has been more significantly eroded, reinforcing the notion of abrasive wear being the dominant wear mechanism.

### 3.4.2. CoCrFeMnNi$_{0.8}$V Coating

Figure 10 shows mass loss as a function of the sliding distance. As in the previous case, the slopes of the curves and the externally applied load lead to the calculation of the wear rate, which is also shown in Table 2.

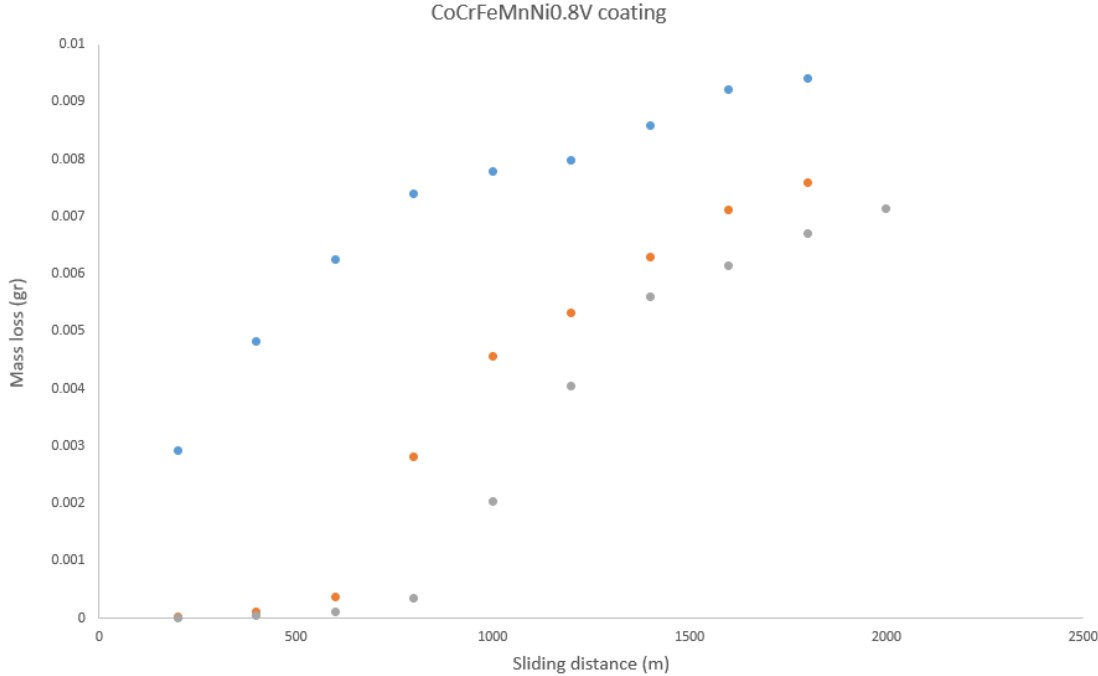

**Figure 10.** Mass loss vs. sliding distance for the CoCrFeMnNi$_{0.8}$V coating.

In Figure 11a–c, the lower magnification SEM image depicts a wear track that is notably even, lacking specific characteristics such as ridges or abrasions that would typically denote abrasive wear. This uniformity in the wear pattern may be indicative of adhesive wear [37]. Upon closer inspection at higher magnification, the surface displays a granulated appearance, with variations in contrast that may indicate the presence of distinct phases or constituents. Brighter regions may represent oxidized sections that have withstood wear, whereas the darker areas may denote locations where the material has been more extensively removed. The microstructure appears largely unaltered, without signs of intense distortion, suggesting that despite the occurrence of wear, the coating has preserved a high degree of its structural integrity.

Energy-dispersive X-ray spectroscopy (EDS) (Figure 11c) analysis offers insights into the elemental distribution along the wear track, with elements such as cobalt (Co), iron (Fe), manganese (Mn), chromium (Cr), and vanadium (V) appearing fairly evenly dispersed. This uniform distribution hints that the wear process impacted the material in a consistent manner, aligning more with a pattern of corrosive or uniform wear, as opposed to localized wear such as pitting or abrasion.

To summarize, the wear exhibited by the CoCrFeMnNi$_{0.8}$V coating is characterized by an evenly distributed material loss without prominent wear markings such as deep channels or scrapes. The EDS findings do not show a selective removal of material, suggesting that the wear process is not predominantly dictated by variations in chemical composition within the coating. The absence of significant cracking or peeling in the images could also imply that fatigue wear is not the primary wear mechanism at play.

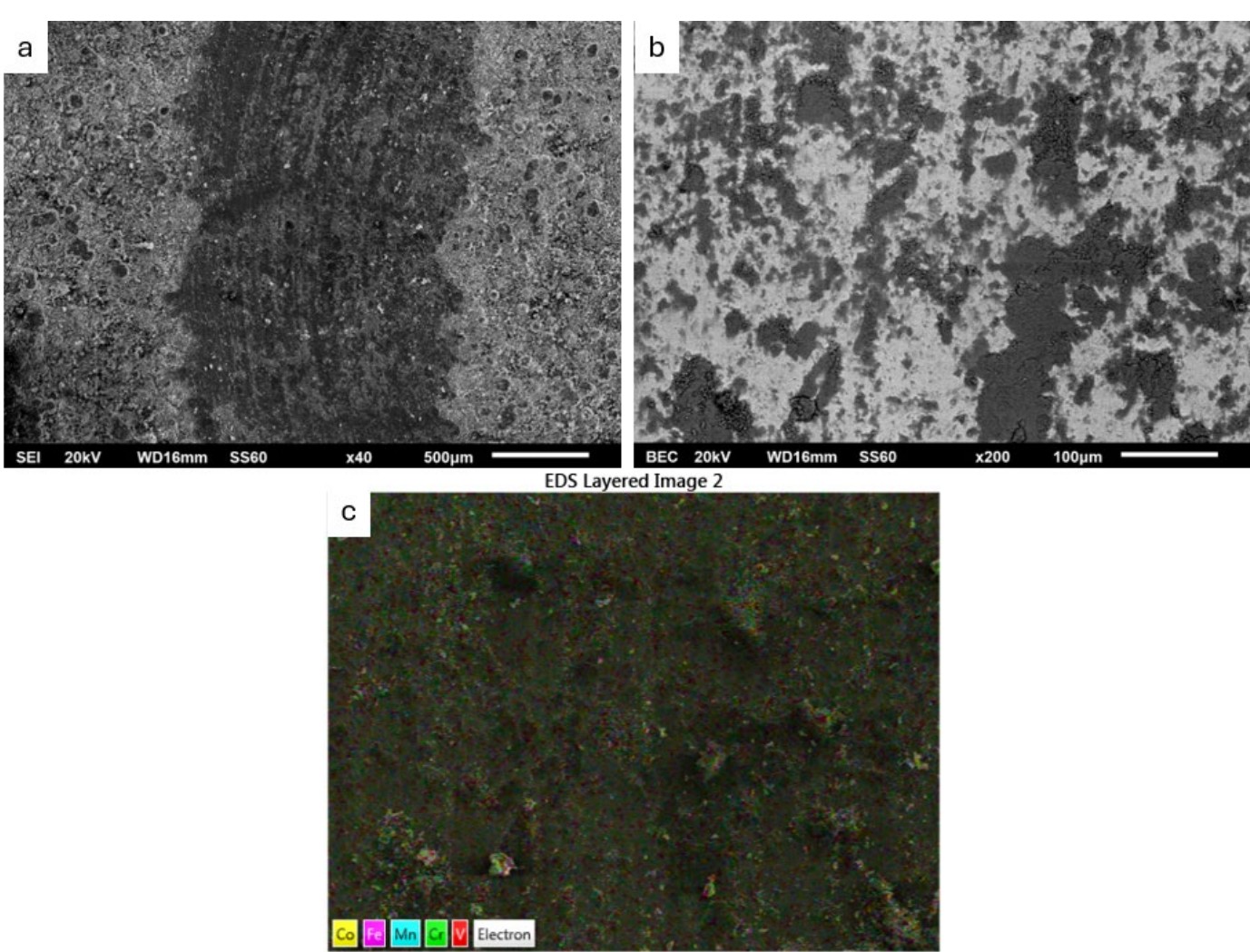

**Figure 11.** Wear tracks of CoCrFeMnNi$_{0.8}$V coating: (**a**) panoramic view, (**b**) higher magnification, (**c**) EDS elemental mapping.

### 3.4.3. CoCrFeMnNi$_{0.8}$V + Cr$_3$C$_2$-Ni$_{20}$Cr Coating

Figure 12 presents the mass loss vs. sliding distance. As in the previous cases, the slopes of the curves and the externally applied load lead to the calculation of the wear rate shown in Table 2.

Figure 13a–d SEM images capture an expansive view of the composite coating surface, where variations in grayscale intensity reveal the existence of different phases or constituents, each with distinct electron densities. Darker regions likely represent worn areas of the CoCrFeMnNi$_{0.8}$V matrix, whereas lighter areas might signify the presence of wear-resistant CrC particles. A closer examination at higher magnification unveils the surface's topography and wear patterns, with the darker areas denoting substantial material loss, possibly due to abrasive- or adhesive-wear mechanisms. The observation of pores and craters indicates that particles have been ejected from the matrix, a phenomenon consistent with both abrasive wear and material fatigue.

Overlaying the EDS analysis on the SEM images maps out the elemental distribution, uniformly highlighting elements like Co, Fe, Mn, Cr, and Ni. The detection of oxygen across the surface points to some level of oxidation, likely resulting from either exposure to high temperatures or corrosive wear. The presence of oxygen alludes to potential oxidation due to wear interactions or environmental factors, while aluminum (Al) traces might originate from external sources or material transfer during the wear test.

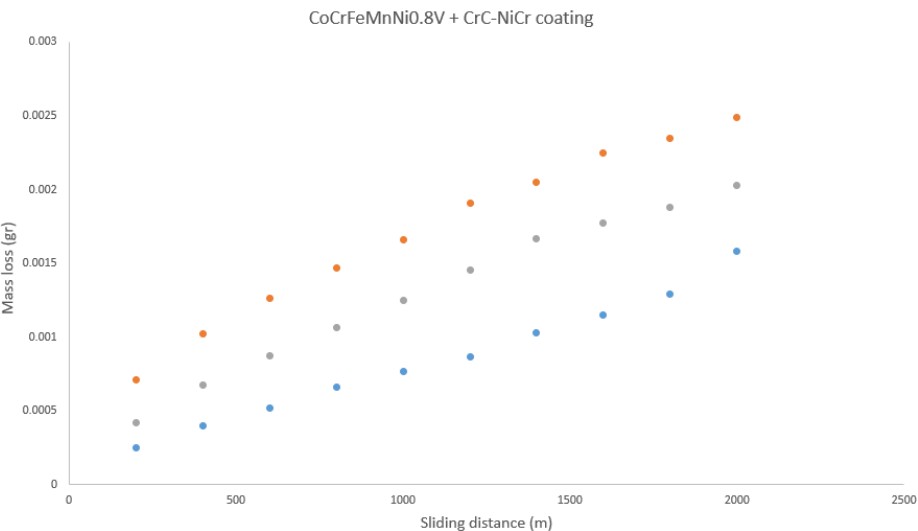

**Figure 12.** Mass loss versus sliding distance for the CoCrFeMnNi$_{0.8}$V + Cr$_3$C$_2$-Ni$_{20}$Cr composite coating.

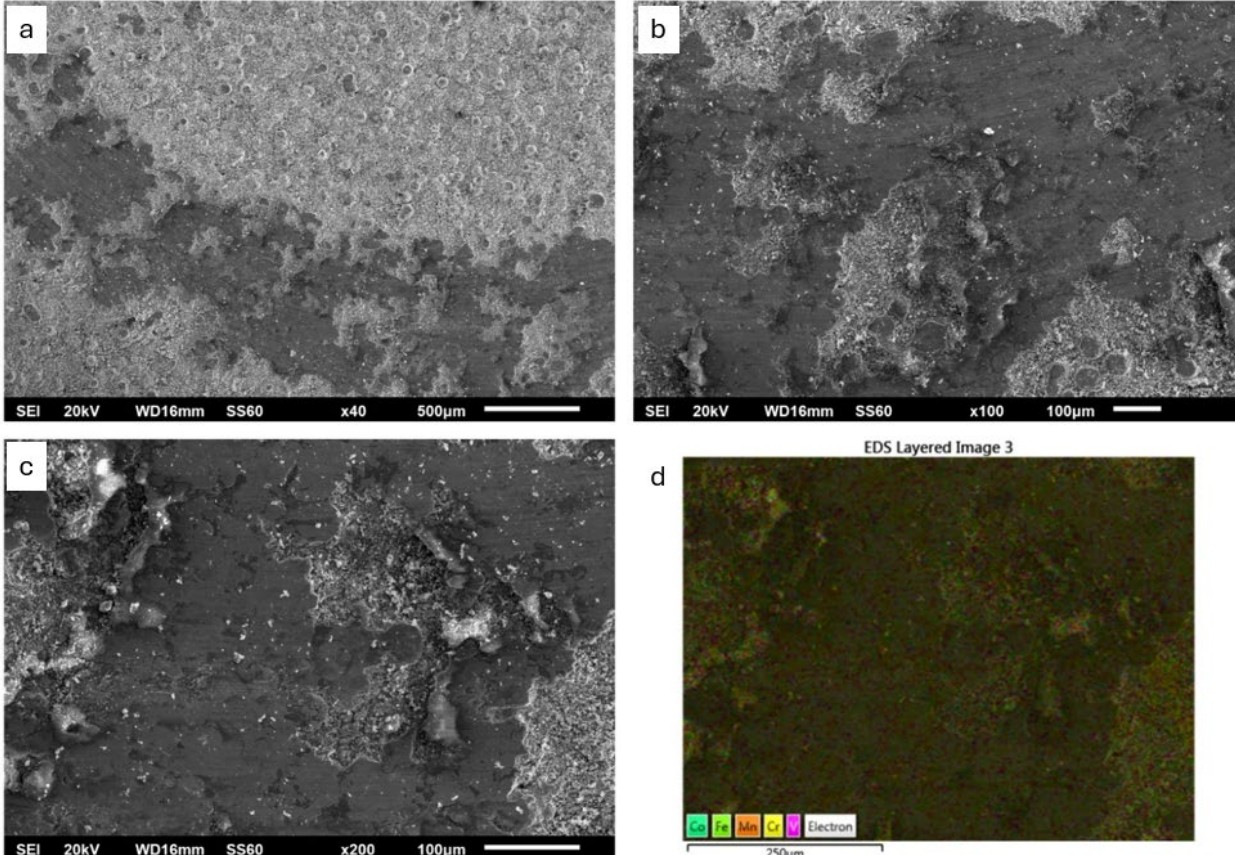

**Figure 13.** Composite-coating wear track at different magnifications (**a**–**c**) and the related EDS chemical mapping (**d**).

In summary, the gathered images and analytical data shown in the previous paragraphs propose a multifaceted wear mechanism involving the following:

1.  Abrasive Wear: evidenced by material removal and the resilience provided by CrC particles, indicating that abrasive actions are at play, with the CrC constituents enhancing the coating's resistance to wear [38–40].
2.  Adhesive Wear: the wear patterns' smoothness and the presence of aluminum hint at material transfer, pointing towards adhesive-wear processes [38,40].

3.  Oxidative Wear: the occurrence of oxygen across the worn surface suggests oxidative wear might also play a role, potentially compromising the matrix's integrity and exacerbating wear effects [38,40].

4.  Fatigue Wear: the observed pores and craters could signal material fatigue, leading to particle detachment, particularly under cyclic-stress conditions [38].

This analysis, therefore, indicates a complex interplay of wear mechanisms impacting especially the monolithic HEA and the reinforced coatings, with contributions from abrasive-, adhesive-, oxidative-, and fatigue-wear processes shaping the observed wear patterns and material loss.

Table 2 presents the average specific wear rates for various coating materials, alongside the primary wear mechanisms for each. The CrC-NiCr coating exhibits the lowest specific wear rate at $1.12 \times 10^{-7}$ g/Nm, demonstrating superior wear resistance. The wear process for this material is identified as a combination of abrasive and fatigue wear. The coating's wear resistance is attributed to the hard carbide particles, which counteract abrasive wear, while the NiCr matrix is susceptible to cyclic stress, leading to fatigue [36–40]. SEM images of this coating reveal carbide particles emerging from the matrix, alongside evidence of NiCr depletion and potential microcracking as signs of fatigue.

Conversely, the $CoCrFeMnNi_{0.8}V$ coating, lacking $Cr_3C_2$-$Ni_{20}Cr$ reinforcement, shows a significantly higher average specific wear rate of $7.13 \times 10^{-7}$ g/Nm. This suggests a complex interplay of wear mechanisms, potentially including abrasive, adhesive, oxidative, and corrosive wear, without a singular dominant process.

The composite material that combines $CoCrFeMnNi_{0.8}V$ HEA with $Cr_3C_2$-$Ni_{20}Cr$ particles achieves an average specific wear rate of $1.7 \times 10^{-7}$ g/Nm. While this rate is higher than that of the pure $Cr_3C_2$-$Ni_{20}Cr$ coating, it is markedly lower than that of the standalone HEA. This indicates that incorporating $Cr_3C_2$-$Ni_{20}Cr$ into the HEA significantly enhances its wear resistance. The observed wear mechanism in the composite material suggests that it benefits from both the $Cr_3C_2$-$Ni_{20}Cr$ particles' hardness and abrasion resistance and the HEA matrix's toughness and capacity for energy absorption. This results in a wear profile that combines the distinct advantages of its components.

The incorporation of $Cr_3C_2$-$Ni_{20}Cr$ into the $CoCrFeMnNi_{0.8}V$ HEA matrix notably diminishes the wear rate in comparison to the pure HEA. This improvement is primarily due to the abrasive wear resistance imparted by the $Cr_3C_2$-$Ni_{20}Cr$ particles, while the HEA matrix enhances the composite's overall toughness and potentially its fatigue-wear resistance. The composite design strategically leverages the individual strengths of its components: the CrC hardness for resisting abrasive forces and the HEA's toughness and alloying complexity for enhanced performance under wear conditions.

### 3.5. Corrosion Resistance

#### 3.5.1. $Cr_3C_2$-$Ni_{20}Cr$ Coating

Figure 14 provide insights into the corrosion behavior of the coating, with the initial low slope following the $E_{corr}$ point indicating an increase in current due to the commencement of active corrosion processes. In two of the examined samples (blue and green), the presence of passivation areas can be observed. This in practice means that, within this specific range of potentials, a stable passivation film is formed, which can obstruct the progress of intensive corrosion phenomena. However, it can also be noticed that these passivation films are not stable; they are dissolved, and hence active corrosion deterioration continues. The current densities for the reverse sequence, which are higher than the forward step, along with the shape of the reverse curves, are also an indication of the instability of the protection passive films; additionally, there is also an indication of localized corrosion having taken place.

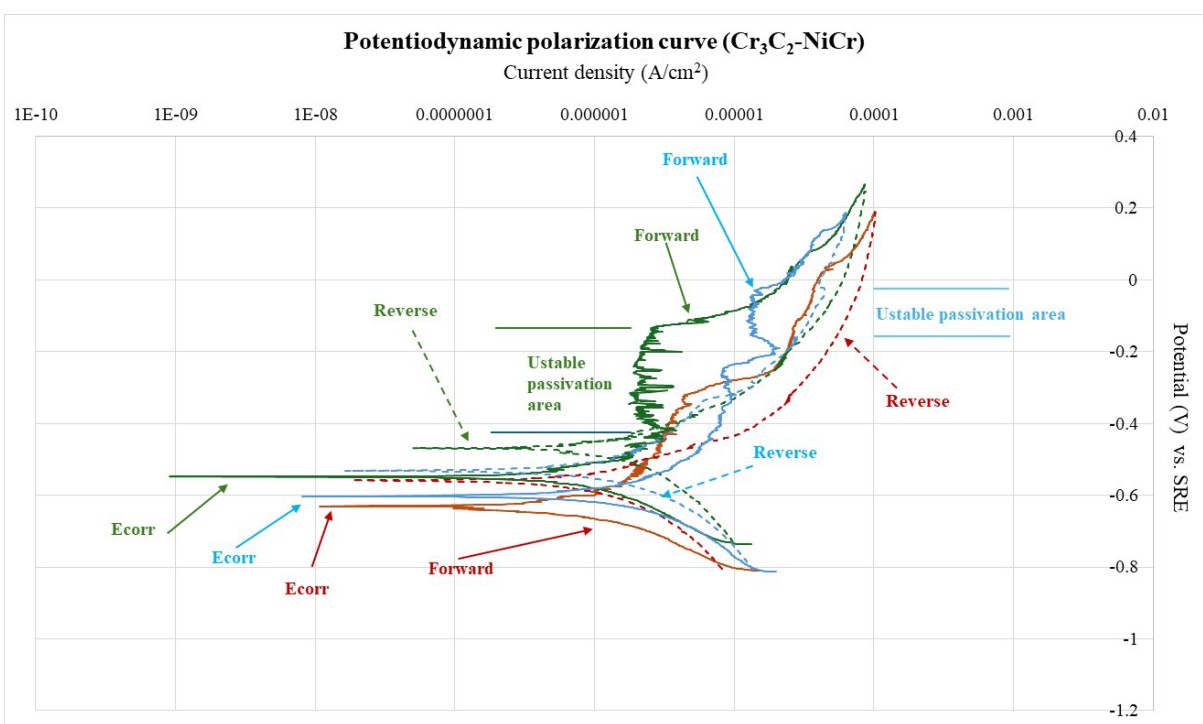

**Figure 14.** Polarization curves of the Cr3C2-Ni20Cr coating.

As shown in Figure 15, SEM/EDS analyses, especially the identified distribution of chlorine and oxygen on the surface, confirm the occurrence of general corrosion on the coating. The preferential concentration of chlorine on the surfaces of carbides indicates that chloride ions from seawater may selectively target the carbide phases or congregate in these regions as a result of electrochemical reactions. This specific interaction highlights the role of chloride ions in influencing the corrosion dynamics, particularly in environments where these ions are prevalent, such as in seawater.

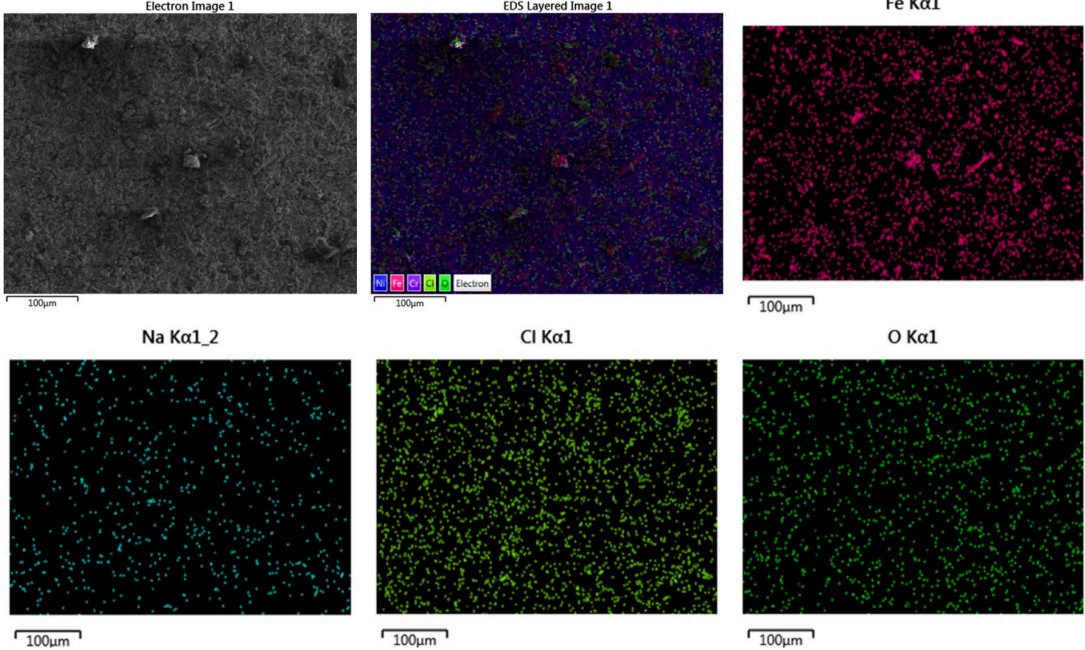

**Figure 15.** SEM/EDS analysis from the corroded surface of the coating.

The $Cr_3C_2$-$Ni_{20}Cr$ coating demonstrates superior corrosion resistance in seawater environments, as evidenced by the low corrosion-current ($I_{corr}$) values, with an average of $8.12 \times 10^{-6}$ A/cm$^2$. This level of resistance is largely attributed to the even, not totally stable, oxide film that forms on the coating's surface, as noticed in Figure 14. The effectiveness of this protective layer is crucial; as long as it remains intact and well-adhered to the substrate, it serves as a formidable barrier against further corrosion, yet in the case of instability, its presence delays the intensive expression of corrosion phenomena. The corrosion resistance of the $Cr_3C_2$-$Ni_{20}Cr$ coating, therefore, hinges on the stability and protectiveness of the oxide film in seawater, which is corroborated by the passive region observed in the polarization curves [41]. The detection of chlorine and oxygen on the coating's surface confirms the occurrence of general corrosion. However, the corrosion-current values indicate that the coating maintains its integrity and continues to offer substantial resistance to further corrosive damage. This characteristic renders the $Cr_3C_2$-$Ni_{20}Cr$ coating an excellent choice for applications demanding high corrosion resistance, particularly in marine settings.

### 3.5.2. CoCrFeMnNi$_{0.8}$V Coating

The polarization curves for the CoCrFeMnNi$_{0.8}$V coating, Figure 16, exhibit relative repeatability, indicating uniform corrosion behavior across the coating. Initially, the forward polarization curve demonstrates an increase in current following the $E_{corr}$ point, signaling the beginning of active corrosion processes. This increase is subsequently followed by a reduced current-increase rate, suggesting the development of protective oxide layers on the coating's surface, which, however, do not show the characteristics of a typical passivation film. The higher current rates during the reverse steps, along with the broad loops observed in the curves, suggest a general form of corrosion [42].

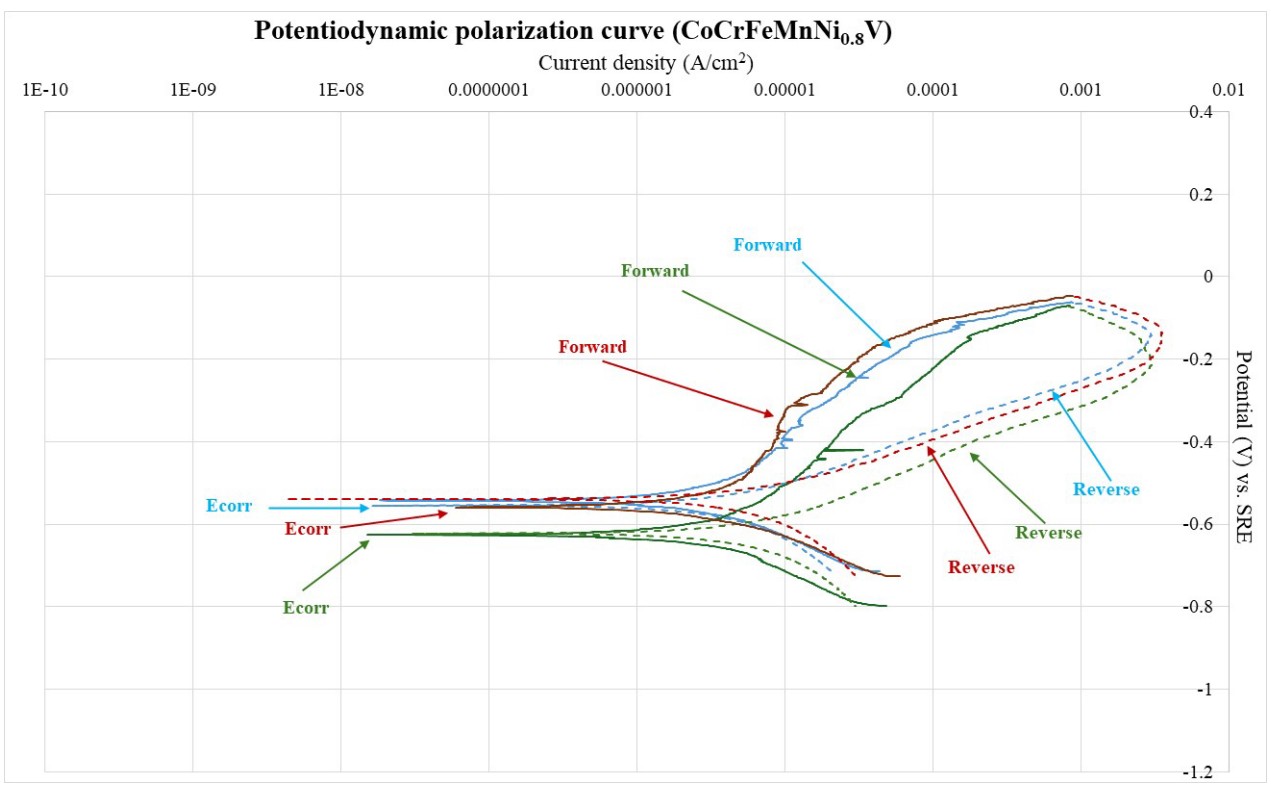

**Figure 16.** Polarization curves of the CoCrFeMnNi$_{0.8}$V coating.

SEM imaging (Figure 17) reveals the occurrence of phase segregation on the coating surface, likely distinguishing between uncorroded and oxidized particles. Areas of the

high-entropy alloy (HEA) that are devoid of chloride highlight the coating's effective corrosion resistance.

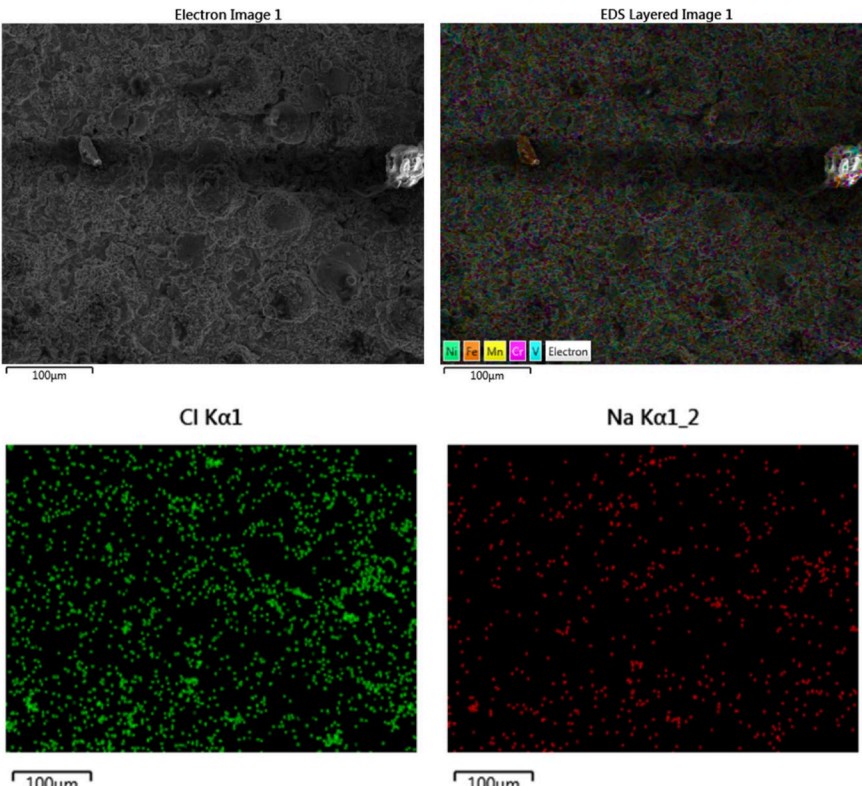

**Figure 17.** SEM/EDS analysis from the corroded surface of the HEA coating.

EDS analysis supports this observation, showing a lack of significant chlorine accumulation on the CoCrFeMnNi$_{0.8}$V particles, as shown in Table 3, further affirming their ability to withstand corrosive attack. However, localized corrosion has been observed, which may be attributed to suboptimal interface quality among unmelted atomized HEA particles. The corrosion-current (I$_{corr}$) values, with an average of $4.06 \times 10^{-6}$ A/cm$^2$, underscore the coating's superior corrosion resistance, especially when compared to Inconel 625, known for its high corrosion resistance, with a reported value of $25.7 \times 10^{-6}$ A/cm$^2$ [42,43].

**Table 3.** Atomic concentration of the elements of the corroded coating.

| Element | Atomic % |
|---------|----------|
| Na | 2.83 |
| Cl | 1.46 |
| V | 14.96 |
| Cr | 17.82 |
| Mn | 15.88 |
| Fe | 17.26 |
| Co | 16.74 |
| Ni | 13.04 |
| Total | 100.0 |

In summary, the CoCrFeMnNi$_{0.8}$V coating shows a considerable capability for resisting corrosion, due to the intrinsic corrosion resistance offered by its HEA composition. Potential enhancements in the coating application process could further augment its performance, establishing the CoCrFeMnNi$_{0.8}$V coating as a highly reliable option for use in marine settings and other environments prone to corrosive damage.

### 3.5.3. CoCrFeMnNi$_{0.8}$V + Cr$_3$C$_2$-Ni$_{20}$Cr Composite Coating

The polarization curves, as shown in Figure 18, show relative repeatability, affirming the consistent corrosion resistance offered by the coating. The polarization curves have the same shape: the forward curve has a high slope, indicating a small current rise and good corrosion resistance. This part of the curve is followed by an area where the current is rapidly raised. Oxides are developed on the surface of the coating and give temporary protection. The developed oxides are not coherent and detach from the coating, causing the current rise. This is a strong indication that the coating presents local corrosion. The reverse curves were shaped at higher currents than the forward, indicating also local corrosion. The third test (green curves) does not present exactly the same curves, but the sample had a similar behavior. The forward curve after the small current area has smaller current values, showing a current decrease. Oxides have been developed on the surface that are not stable and after a while they detach, causing a rise in the current, indicating also local corrosion.

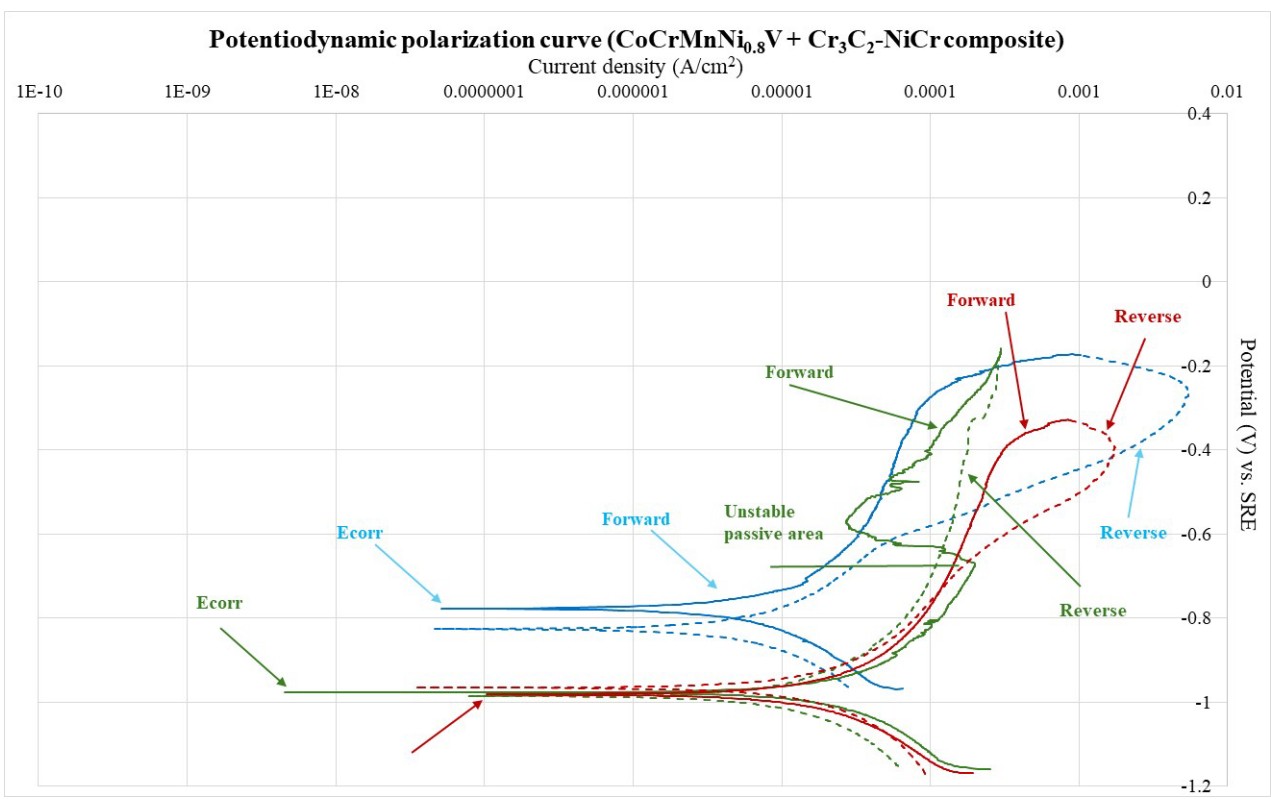

**Figure 18.** Polarization curves of the composite coating.

The relatively low I$_{corr}$ values (an average of 9.13 $\times$ 10$^{-6}$ A/cm$^2$), especially when compared to those of Inconel 625, highlight the coating's significant corrosion resistance in seawater conditions [41–43]. SEM imaging (Figure 19) reveals phase segregation on the surface, likely resulting from unmelted particles and the development of oxide layers. EDS analyses further confirm the oxide presence and indicate chlorine's attack on Cr$_3$C$_2$-Ni$_{20}$Cr, while CoCrFeMnNi$_{0.8}$V particles exhibit no chlorine traces, underscoring their increased resistance to corrosion.

Localized corrosion is suspected to arise from suboptimal interfaces between HEA particles and carbides, as suggested by microstructure examination. SEM-EDS point analysis indicates that the observed segregation and oxide formations on the surface may play a role in the corrosion dynamics, particularly in areas exhibiting localized corrosion. This analysis underscores the complex interplay between material composition, microstructural features, and corrosion resistance, highlighting the need for careful consideration of mate-

rial interfaces and oxide formation in enhancing the overall corrosion resistance of coatings in marine and other corrosive environments.

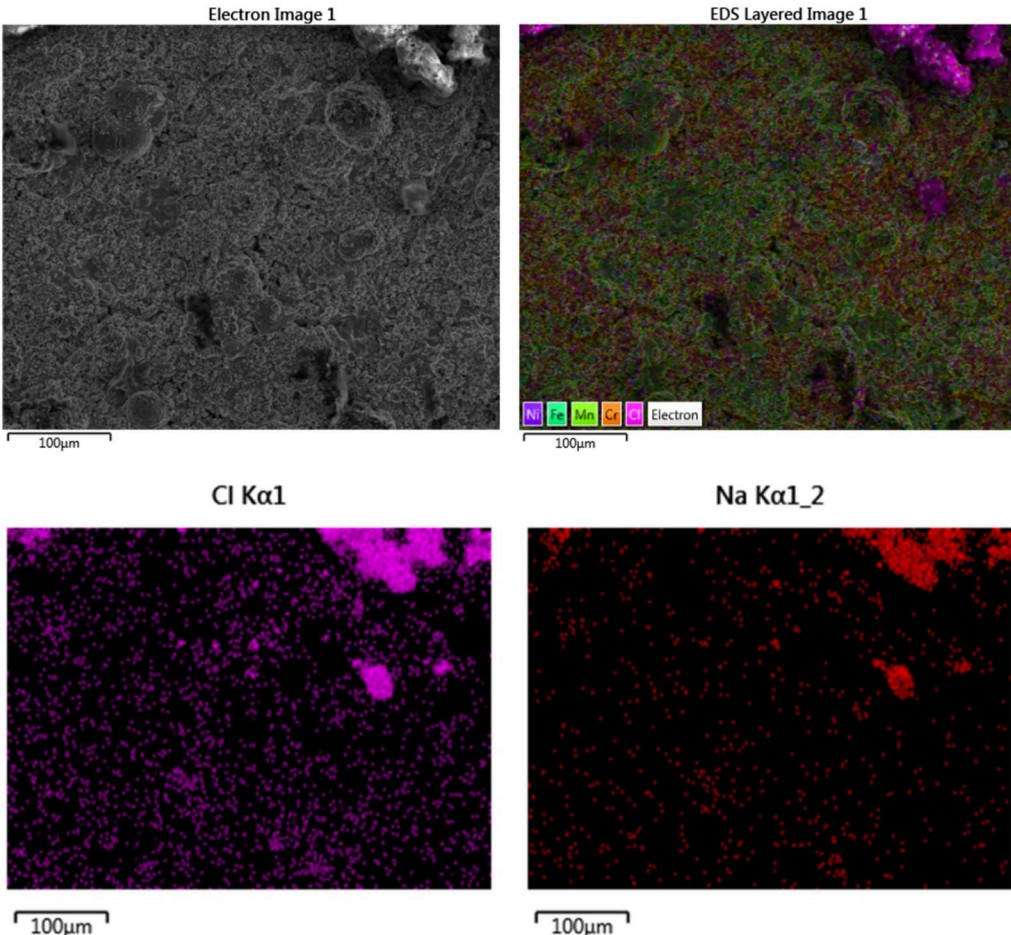

**Figure 19.** SEM/EDS analysis from the corroded surface of the composite coating.

The composite coating, comprising 75% $CoCrFeMnNi_{0.8}V$ and 25% $Cr_3C_2$-$Ni_{20}Cr$, exhibits an optimum combination of wear and corrosion resistance, rendering it well-suited for challenging conditions such as seawater exposure. The formation of coherent oxides on the coating surface indicates a corrosion-resistance mechanism characterized by the establishment of a protective barrier, effectively reducing further material deterioration.

In summary, the $CoCrFeMnNi_{0.8}V$ + $Cr_3C_2$-$Ni_{20}Cr$ composite coating, as in the case of the other two systems tested in the present effort, showed a better corrosion resistance compared to high-performance materials like Inconel 625. It holds promise for enhanced durability and efficacy through further refinement of the coating application process. The coating's resilience against corrosion in maritime settings is significantly influenced by its microstructural characteristics, the distribution of different phases within the composite, and the nature of oxide layers that form on its surface.

As summarized in Table 4, the $CoCrFeMnNi_{0.8}V$ coating exhibits the lowest corrosion-current density ($I_{corr}$), signifying superior corrosion resistance among the tested coatings, along with its higher $E_{corr}$ compared to the composite. This indicates that the pure high-entropy alloy (HEA) coating offers the best protection against corrosion. The composite system, on the other hand, exhibited the worst behavior compared to the other two systems. It has to be mentioned, however, that its performance is close to that of the reinforcing phase. It could be postulated that the introduction of the reinforcing phase did not improve the superior behavior of the monolithic HEA coating. This is most likely due to the fact that this introduction of the reinforcing phase in the HEA matrix in the coating form creates

more interfaces and potential areas of weakness (pores, microflaws, etc.), which could contribute to this negative behavior. Nevertheless, the corrosion response of the composite coating should not be examined independently of the overall behavior. The fact that the corrosion response is not very deteriorated, especially compared to that of the reinforcing phase, in conjunction with its advanced wear resistance, indicates that as a potential coating system it does possesses optimum overall performance.

**Table 4.** Polarization Testing: Comparative Summary.

| Coating System | $E_{corr}$ (mV) | $I_{corr}$ ($\times 10^{-6}$) A/cm$^2$ | Remarks |
|---|---|---|---|
| CoCrFeMnNi$_{0.8}$V | −302.00 | 4.06 | Local corrosion. Very low Icorr. |
| CoCrFeMnNi$_{0.8}$V + CrC-NiCr (75–25)% Mix | −462.00 | 9.73 | Local corrosion. Very low I$_{corr}$. |
| CrC-NiCr | −359.00 | 8.12 | General corrosion. Passivation. |

### 3.5.4. Summary

The Cr$_3$C$_2$-Ni$_{20}$Cr coating demonstrates a passive behavior in its polarization curve, indicative of a protective oxide film's formation on its surface. This film is vital for enhancing the coating's corrosion resistance. SEM/EDS analysis reveals widespread corrosion across the surface, with chlorine and oxygen particularly concentrated on carbide regions. This pattern suggests a selective corrosion process, likely driven by chloride ions targeting the carbide areas.

For the CoCrFeMnNi$_{0.8}$V coating, polarization data show a reduction in current following initial oxidation, pointing towards the formation of a protective oxide layer that contributes to its effective corrosion resistance. SEM/EDS analysis displays some degree of phase segregation; however, a notably low concentration of chlorine on the alloy particles suggests a robust resistance against chloride ion corrosion.

The composite coating exhibits a balanced corrosion resistance, though it shows potential for localized corrosion, possibly due to suboptimal interfaces between HEA particles and carbides. SEM/EDS analysis indicates phase segregation and the presence of oxide layers, with the Cr$_3$C$_2$-Ni$_{20}$Cr components appearing more susceptible to chlorine attack in comparison to the CoCrFeMnNi$_{0.8}$V particles. This analysis highlights the complex interplay between material composition, microstructure, and corrosion resistance, underscoring the importance of optimizing interface quality to enhance the overall performance of composite coatings in corrosive environments.

## 4. Conclusions

The composite coating, blending the high-entropy alloy (HEA) CoCrFeMnNi$_{0.8}$V with Cr$_3$C$_2$-Ni$_{20}$Cr, leverages the advantages of both constituents to offer superior protection against corrosion, notably in environments abundant in chloride ions, like seawater. This synergy results in the formation of a protective oxide layer, pivotal in hindering further degradation of the material.

The dual nature of the coating achieves a harmonious balance between wear and corrosion resistance. The Cr$_3$C$_2$-Ni$_{20}$Cr component enhances wear resistance, while the CoCrFeMnNi$_{0.8}$V boosts corrosion resistance. This combination ensures the coating's suitability for demanding applications exposed to aggressive conditions.

In summary, the composite HEA + Cr$_3$C$_2$-Ni$_{20}$Cr coating emerges as a highly effective solution for scenarios that demand both exceptional corrosion resistance and durability, particularly in marine settings. This innovative blend of HEA's corrosion resistance and Cr$_3$C$_2$-Ni$_{20}$Cr's wear resistance marks a significant progression in the field of protective coatings, presenting a valuable option for extending the life and performance of materials in challenging environments.

**Author Contributions:** Conceptualization, S.K. (Spyros Kamnis), A.E.K. and E.G.; methodology, S.K. (Spyros Kamnis), S.K. (Stavros Kiape), A.E.K., E.G., M.G. and T.E.M.; validation, S.K. (Spyros Kamnis), S.K. (Stavros Kiape), A.E.K., E.G. and M.G.; formal analysis, S.K. (Spyros Kamnis), S.K. (Stavros Kiape), A.E.K. and E.G.; investigation, S.K. (Stavros Kiape), M.G. and S.K. (Spyros Kamnis); resources, S.K. (Spyros Kamnis), A.E.K. and T.E.M.; data curation, S.K. (Stavros Kiape), A.E.K. and E.G.; writing—original draft preparation, S.K. (Stavros Kiape), S.K. (Spyros Kamnis) and A.E.K.; writing—review and editing, S.K. (Stavros Kiape), S.K. (Spyros Kamnis) and A.E.K.; visualization, S.K. (Stavros Kiape) and S.K. (Spyros Kamnis); supervision, A.E.K., E.G. and S.K. (Spyros Kamnis). All authors have read and agreed to the published version of the manuscript.

**Funding:** This research received no external funding.

**Institutional Review Board Statement:** Not applicable.

**Informed Consent Statement:** Not applicable.

**Data Availability Statement:** Data will be available at demand.

**Conflicts of Interest:** The authors would like to declare no conflicts of interest.

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
