# Peer review of "CoCrFeMnNi0.8V/Cr3C2-Ni20Cr High-Entropy Alloy Composite Thermal Spray Coating: Comparison with Monolithic CoCrFeMnNi0.8V and Cr3C2-Ni20Cr Coatings"

_coatings, doi:10.3390/coatings14040402_

Round 1

Reviewer 1 Report

Comments and Suggestions for Authors

The following minor comments need to be addressed.

1. The contributions need to be added at the end of introduction.

2. The motive of merging HEAs with CrC-NiCr need to added.

3. In Section 2.7, the authors mentioned "while maintaining a neutral pH.". However, no results related to this.

4. Detailed analysis from Polarization curves have to be added.

5. There are several mistakes in nomenclature. For instance, Icorr or Icorr two different notations.

6. Material characteristics such as melting temperature, density, composition weight need to added.

7.  Add more details on corrosion performance.

Comments on the Quality of English Language

Fine

Author Response

Dear Reviewer,

We would like to thank you for your important comments and suggestions in an effort to improve the quality of our manuscript. In the following point we try to respond in each individual of the proposed remarks.

  1. The contributions need to be added at the end of introduction.

Response: The authors are not entirely sure what exactly is the suggestion of the reviewer. If the reviewer means the contribution of each author into the overall research of this work and the preparation of the manuscript, we think that this is something that is generated automatically by the submission platform. If the reviewer means the general contribution of the work in the field, then the scope is mentioned at the end of subsections 1.1, 1.2 and 1.3 of the introduction. We would really appreciate if the reviewer could assist us towards this direction.

  1. The motive of merging HEAs with CrC-NiCr need to added.

Response: A paragraph was added in the revised manuscript at the end of sub section 1.2 of the introduction.

  1. In Section 2.7, the authors mentioned "while maintaining a neutral pH.". However, no results related to this.

Response: Further explanation of the pH treatment was been added on the experimental section 2.7.

  1. Detailed analysis from Polarization curves have to be added.

Response: The polarization curves have been entirely modified.  

  1. There are several mistakes in nomenclature. For instance, Icorr or Icorr two different notations.

Response: These mistakes were traced and corrected as suggested.

  1. Material characteristics such as melting temperature, density, composition weight need to added.

Response: Some properties of the individual components have been added:

HEA: density 7.68 m/cm3, melting point 1573 oC (caution: these values are calculated values based on basic rule of mixtures)

Cr3C2: density 6.68 g/cm3, melting point 1622 oC

Ni20Cr (%at.) : density 7.49 g/cm3, melting point 1440 oC

The reinforcing phase is a mixture of Cr3C2 -Ni20Cr (75-25 in wt%)

The composite is a mixture of HEA-reinforcing phase (75-25 in wt%)

  1. Add more details on corrosion performance.

Response: The corrosion session has been completely altered.

Please refer to the attached file

Reviewer 2 Report

Comments and Suggestions for Authors

This contribution is important but requires some additional work as indicated in the attached annotated manuscript. The paper is rather verbose and in parts repetitive, in particular when expounding the apparent advantages of the HEA coatings. Despite evidence to the contrary, the authors maintain a kind of sef-fulfilling prophesy: they gather from the literature that HEA may be excellent contenders for a variety of demanding applications and then set out to prove that composite HEACrC/NiCr coatings are superior which they are not. In particular, the results of the potentiodynamic studies are ambiguous and may even contain errors. The authors are requsted to study carefully the attached annotated manuscript and comply with the suggestions of this reviewer.

Round 2

Reviewer 2 Report

Comments and Suggestions for Authors

The authors revised successfully their contribution along the suggestions of this reviewer. All queries and critical comments were properly addressed by the authors.